# Rubella virus IgM and IgG antibodies with avidity in pregnant women and outcomes at a tertiary facility in Ghana

**Naa Baake Armah[1]\*, Kwamena W. Sagoe[2]\*, Mercy Nuamah[1], Alfred E. Yawson[3], Edmund T. Nartey[4], Vera A. Essuman[5], Nana-Akyaa Yao[6], Kenneth K. Baidoo[5], Jemima Anowa Fynn[5], Derrick Tetteh[2], Eva Gyamaa-Yeboah[1], Makafui Seshie[2], Isaac Boamah[2], Kobina Nkyekyer[1]**

1 Department of Obstetrics and Gynaecology, Korle Bu Teaching Hospital, Accra, Ghana, 2 Department of Medical Microbiology, University of Ghana Medical School, College of Health Sciences, University of Ghana, Legon, Ghana, 3 Department of Community Health, University of Ghana Medical School, College of Health Sciences, University of Ghana, Legon, Ghana, 4 Center for Tropical Clinical Pharmacology and Therapeutics, University of Ghana Medical School, College of Health Sciences, University of Ghana, Legon, Ghana, 5 Department of Surgery, University of Ghana Medical School, University of Ghana, Accra, Ghana, 6 Cardiothoracic Centre, Korle Bu Teaching Hospital, Accra, Ghana

\* nbarmah2@yahoo.com (NBA); kwsagoe@ug.edu.gh (KWS)

## Abstract

### Background

Congenital rubella syndrome (CRS) is a recognised cause of childhood deafness and blindness caused by the transplacental transmission of rubella virus during pregnancy. Women in the reproductive age group, and by extension their unborn babies may therefore be at increased risk. The prevalence of Rubella virus specific IgM and IgG antibodies, including IgG avidity, was determined in pregnant women attending the antenatal clinic at a Teaching Hospital in Ghana.

### Methods

One hundred and forty-five women in their second and third trimesters of pregnancy from the outpatient clinic were recruited over a period of 2 months after written informed consent was obtained. Study participants completed a questionnaire and venous blood drawn for IgM, IgG, and avidity testing using SERION ELISA (SERION® Immunologics, Würzburg, Germany). Babies of mothers with positive or indeterminate IgM and low avidity IgG antibodies were offered specialist cardiological, ophthalmological or hearing assessment during follow up.

### Results

One hundred and twenty-eight (88.3%) had only IgG antibodies, 5 (3.4%) had IgM and IgG antibodies, while 12 (8.3%) had no antibodies. No patient had IgM antibodies alone. Ten women (6.9%) had indeterminate levels of IgM antibodies. Majority of the women had high avidity IgG antibodies, while 5 (3.4%) had low avidity antibodies. No patient had IgM with

**Data Availability Statement:** All relevant data are within the paper.

**Funding:** The authors received no specific funding for this work.

**Competing interests:** The authors have declared that no competing interests exist.

low avidity antibodies. There was no statistical association between socio-demographic factors and the presence of IgM, IgG (low or high avidity) antibodies. Of all the children followed, none had the clinical definition of CRS.

## Conclusions

Consistent with the World Health Organization elimination strategy for measles and rubella viruses, non-immune women in the reproductive age group should be vaccinated. The immunization programme should be expanded to include teenagers and adults. Though Congenital Rubella Syndrome was not detected, the risk still remains.

## Introduction

Mathematical models estimate the number of congenital rubella syndrome (CRS) cases in developing countries to be more than 110,000 per annum with majority of the disease burden in Asia (48%) and Africa (38%) [1], and majority of these are in developing countries without rubella vaccination programmes. The highest incidence of these defects have been associated with the first trimester and subsequently reduce as gestational age advances [2]. The World Health Organization (WHO) has therefore suggested the antenatal screening of pregnant women as one of the arms for rubella surveillance [3]. The burden of CRS in Ghana has been previously described [4].

Global rubella virus seropositivity estimates varies with WHO regions in pregnant women and women of child bearing age, with susceptible rates of > 5% [5]. Several studies from West Africa using ELISAs from different companies have recorded varying prevalence rates of rubella virus (RV) specific IgM (1.36–38.8%) and IgG (53.0%–95.7%) [6–14]. There are always varying percentages of non-immune pregnant women, and most of these studies did not identify specific risk factors. Avidity testing provides information on the recency of RV specific IgG. Long standing infections have higher avidities as compared with recent infections [15]. In all the aforementioned studies, none performed avidity testing. The avidity testing seems to enhance the detection of recent infections as compared to re-infections [15]. To our knowledge, there is little data on the use of avidity testing for RV IgG [16].

The Ghana Health Service states that there are over 700 cases of CRS in Ghana every year, with a stated IgG seroprevalence in pregnant women of 92.6% (Personal Communication). In 2013 the Ghana Health Service through GAVI and WHO introduced the combined measles and rubella vaccine as part of its expanded programme of immunization. The vaccine was introduced throughout the country with special mass vaccination campaigns targeting children between the ages of 9–14 years as well in a successful start in the three northern regions (http://www.gov.gh/). However, vaccination strategies that focused on women and children in Brazil were associated with a 5.5-fold increase in rubella in men and an increase in the incidence of CRS [17]. More recent data in Ghana suggests that the non-immune status of pregnant women may be >5% [18, 19]. There is therefore likely to be a significant risk for CRS in Ghana.

The aim of this study was therefore to determine the seroprevalence of rubella virus antibodies including avidity testing in women attending antenatal care at the Korle Bu Teaching Hospital, Accra, and the prevalence of CRS in babies born to mothers with RV IgM.

## Materials and methods

### Study site and area

The study was carried out at the antenatal clinic of the Department of Obstetrics and Gynaecology of the Korle Bu Teaching Hospital, Accra. Babies born to mothers with RV specific IgM were reviewed at the specialty clinics of the Paediatric, Ophthalmology and Otorhinolaryngology departments. The hospital is the largest tertiary facility in Ghana and a major referral centre for southern Ghana with a total of approximately 10,000 deliveries a year. About 40 new patients are seen for antenatal care daily, 70% of whom are in their second or third trimesters. The women come from a wide spectrum of socioeconomic, religious and cultural backgrounds.

### Study design and population

The study was a prospective study (single cohort) involving a group of pregnant women i.e., all women who consented to be screened for RV IgM and IgG antibodies. Patient recruitment was conducted from December 10, 2019 to January 30, 2020 and follow-up of babies done till September 11, 2020. The study included clients attending antenatal care at the outpatient clinic of the obstetric unit. Twenty-five (25) questionnaires were pretested to identify problems and rectified. Women in their second or third trimester of their pregnancies were invited on consecutive clinic days to participate in the study. For those who gave their consent to participate, the questionnaires were administered to obtain sociodemographic characteristics and obstetric history.

### Sample size and strategy

The appropriate sample size for this study was determined by three factors:

i) the estimated prevalence of CRS in Kumasi [4] i.e., 0.8/1000, ii) the desired level of confidence, and iii) the acceptable margin of error.

Using the Cochran formula:

$$n = z^2 \text{ x p } (1-p)/m^2,$$

where

n = required sample size
z = confidence level of 95% (standard value of 1.96)
p = the estimated prevalence of CRS in Kumasi in 2000 (0.8%) [4]
m = margin of error at 5% (standard value of 0.05)

The estimated sample size for the study is 113. Accounting for such contingencies as non—response or refusals, the sample is increased by 20% for an overall minimum sample size of 135.

Consenting clients between the age of ≥18 years in their second and third trimesters (14 to 42 weeks), were included in the study. Women with documented or known history of any autoimmune disease or who declined to participate were excluded from the study.

The approximate number of eligible new cases each day was 30. Majority of antenatal attendants were seen in five consulting rooms from Monday to Friday. At each clinic day, the first patient was selected from the first six patients by the throw of a dice (n) at the reception area. Every clinic day, three to four patients were selected. The subsequent patients selected were every n[th] patient after that, depending on the actual number of patients present at the time the dice was thrown. If a patient declined to participate in the study, the next patient following in the order will be selected.

## Data collection

Structured questionnaires were administered to gather sociodemographic information on age, place of residence, marital status, level of education, occupation and level of income, and religious background. Current obstetric and past medical history including history of vaccination with rubella virus, history of a rash in current pregnancy, outcomes of past deliveries, and the approximate developmental assessment of any previous children, and whether she had any children with a documented visual or hearing defect, were obtained.

## Testing for rubella virus specific antibodies and avidity

Blood samples were collected using EDTA anticoagulant tubes and stored at 4–8˚C for a maximum of 6 hours after which plasma was separated by centrifuging at 1000 x g for 10 minutes. The plasma was aspirated transferred aseptically to a sterile labelled vials with the clients' details. The plasma was stored at 4–8˚C for up to 7 days and kept at– 20˚C if longer storage was desired.

Presence of RV specific immunoglobulins and avidity in pregnant women; IgM, IgG and IgG avidity testing, was done with a quantitative ELISA classic rubella virus IgG/IgM assay, SERION ELISA (SERION® Immunologics, Würzburg, Germany), according to manufacturer's instructions. This assay had been previously used to determine the avidity of RV specific IgG in a study with pregnant women in Nigeria [16].

RV specific IgM antibodies are usually no longer detectable after 2 months while IgG persists [20]. Using the gestational age at which the blood samples were taken, six weeks was subtracted and the gestational age at which the infection occurred was estimated.

## Detection of congenital rubella syndrome

All clients who participated in the study were contacted within one month of sample collection on which to collect reports and have discussion on the interpretation of the reports. Patients who had been exposed to RV during the pregnancy were counselled as to what it means for the unborn infant. Those who qualified were offered anomaly ultrasound scans by a consultant obstetrician. They were placed in contact with the hearing assessment unit, the paediatric eye clinic and the clinical psychology unit of the Korle Bu Teaching Hospital. The hearing assessment unit of the hospital runs a new-born hearing assessment service while the paediatric eye clinic runs a programme to identify and manage paediatric cataract. The paediatric cardiologist was informed and the new born had echocardiographic assessment of the heart at the National Cardiothoracic Centre.

The criteria for the final classification of CRS was determined in part by identifying Group A or Group B clinical signs of CRS [21]: i) Cataract(s), congenital glaucoma, pigmentary retinopathy, congenital heart disease (most commonly peripheral pulmonary artery stenosis, patent ductus arteriosus or ventricular septal defects), hearing impairment.

ii) Purpura, splenomegaly, microcephaly, developmental delay, meningoencephalitis, radiolucent bone disease, jaundice that begins within the first 24 hours after birth.

Laboratory testing for CRS was done by taking one millilitre of blood sample from the new born for IgM antibody detection. Baseline full blood count and liver function tests were also done. A diagnosis of laboratory confirmed CRS was considered if a new born had at least one sign from Group A with detection of IgM in the blood sample [21].

## Data management and analyses

All statistical analysis was done with the SPSS Statistics for Windows, version 20.0 (SPSS Inc., Chicago, Ill., USA). Categorical data sets were compared using the chi squared test and where

appropriate the odds Ratio (OR) with 95% confidence interval was calculated. The odds ratio is used to evaluate the risk profile of non-immune women with regard to maternal age, parity and gestational age. The student *t* test is used to compare the ages, gestational ages and avidity index of immune and non-immune patient. Differences are considered significant if the p value is less than 0.05.

## Ethics

Approval for the study was obtained from the Institutional Review Board of the Korle-Bu Teaching Hospital (KBTH-IRB 00092/2018). The study was explained to the prospective participants and informed written consent obtained and witnessed. All data was handled confidentially.

## Results

A total of 145 women were enrolled for the study over a 2-month period. Of this number, 133 (91.7%) had been exposed to RV and 12 (8.3%) were non-immune as determined by the presence or absence of specific rubella virus antibodies, respectively.

### Sociodemographic characteristics of study population

The average age of the patients was 31.6 ±5.8 years with most participants living in urban areas (136, 93.8%). Most of the women (112, 77.2%) were married and had been educated beyond the JHS level. Majority (93, 64.1%) earned less than GHC900 per month (Table 1).

The mean number of children each participant had (parity) was 2.1 ±1.2. Sixty -five percent had used a family planning method before, with the male condom (35, 37.2%) and calendar method (26, 27.7%) being the most popular. However, only 9 (9.6%) had used the implant before. The history of a non-itchy, non-discharging rash was not common; it was reported by only 10 (6.9%) respondents. Only a few respondents had heard of rubella before (9, 6.2%). One person (0.7%) had been vaccinated against rubella (Table 2).

### Prevalence of IgM, IgG, and low avidity

A total of 133 (91.7%) were positive for IgG only and 5 (3.4%) were positive for both IgG and IgM. Twelve women (8.3%) did not show serological evidence of exposure to RV (Table 3). A high avidity index was found in 128 (88.3%) of the women. There were 5 women who had IgG antibodies only with low avidity.

### Risk factors for presence of IgG, IgM and low avidity

There was no statistically significant association between age, parity, area of residence, and educational level and the presence of RV IgG or IgM (Tables 4 & 5). When comparing the area of residence for IgM positivity, the p-value was significant for rural areas (OR = 31, p = 0.022). However, the CI of [1.63–589.45] was very wide due to the small sample size. There was also no association between age, parity, area of residence, and educational level and RV IgG low and high avidity (Table 5).

### Estimation of date of infection

The five women who were RV IgM positive, and had low avidity IgG, all lived in urban areas. Data for Client 095 was incomplete and therefore the gestational age at the time of testing and estimated time of infection with rubella could not be calculated (Table 6). The client could also not be reached on any of the contact numbers for follow up. Client 166 was of interest, as the

**Table 1. Socio-demographic characteristics of study participants.**

| Characteristic | Frequency | Proportion (%) |
|---|---|---|
| Age, years (mean ± SD) | 31.8 ± 5.7 | |
| Type of residence | | |
| Rural | 2 | 1.4 |
| Slum | 1 | 0.7 |
| Peri-urban | 6 | 4.1 |
| Urban | 136 | 93.8 |
| Occupation | | |
| Unemployed | 9 | 6.2 |
| Artisan | 35 | 24.1 |
| Trader | 48 | 33.1 |
| Businesswoman | 4 | 2.8 |
| Professional | 42 | 29.0 |
| Civil/Public servant | 5 | 3.5 |
| Housewife | 2 | 1.4 |
| Marital status | | |
| Single | 21 | 14.5 |
| Married | 113 | 77.9 |
| Co-habitation | 11 | 7.6 |
| Educational level | | |
| None | 6 | 4.1 |
| Islamic | 1 | 0.7 |
| Primary | 7 | 4.8 |
| JHS/Middle school | 43 | 29.7 |
| SHS/Technical/Vocational | 34 | 23.5 |
| Tertiary | 54 | 37.2 |
| Estimated monthly income (GhCedis) | | |
| ≤300 | 52 | 43.3 |
| 310–600 | 30 | 25.0 |
| 601–900 | 11 | 9.2 |
| 901–1200 | 10 | 8.3 |
| 1201–1500 | 4 | 3.3 |
| 1501–1800 | 3 | 2.5 |
| 1801–2100 | 2 | 1.7 |
| >2100 | 8 | 6.7 |

SD = Standard deviation; JHS = Junior high school; SSS = Senior high school.

anomaly scan showed multiple congenital anomalies, hydrocephalus and polycyctic kidneys, none of which was consistent with congenital rubella syndrome. The women likely had exposure to RV in late second or early third trimester and therefore stood a very low risk of CRS in their babies. Five clients were positive for IgM with high avidity IgG, suggesting an infection took place at least 13 weeks before the sample was taken.

The average parity was 1.4 and the average gestational age was 30 weeks. Four lived in urban areas and one (036) in a semi urban area. Two clients declined follow up on account of not wanting to frequent a hospital in this period, when their children were apparently well. One client could initially not be contacted until 7 months after delivery as her phone had developed a problem. Only two clients 161 and 168 have completed follow up for their babies.

**Table 2. General obstetric characteristics of study participants.**

| Characteristic | Frequency | Proportion (%) |
|---|---|---|
| Parity | | |
| Parous | 112 | 77.2 |
| Nulliparous | 33 | 22.8 |
| Number of children | | |
| None | 33 | 22.8 |
| 1 | 44 | 30.3 |
| 2 | 31 | 21.4 |
| 3 | 22 | 15.2 |
| $\geq 4$ | 15 | 10.3 |
| Number of children (N = 112) (mean ± SD) | 2.1 ± 1.2 | |
| Ever used a family planning method | | |
| Yes | 94 | 64.8 |
| No | 51 | 35.2 |
| Family planning method used (N = 94) | | |
| Male condom | 35 | 37.2 |
| Calendar method | 26 | 27.7 |
| Oral conceptive pill | 17 | 18.1 |
| Injectable contraceptive (3 monthly) | 16 | 17.0 |
| Female condom | 13 | 13.8 |
| Implant | 9 | 9.6 |
| Intrauterine contraceptive device | 8 | 8.5 |
| Injectable contraceptive (2 monthly) | 2 | 2.1 |
| Non-itchy, non-discharging rash on face/body | | |
| Yes | 10 | 6.9 |
| No | 134 | 93.1 |
| Duration of non-itchy, non-discharging rash (N = 10) | | |
| 1 month | 2 | 18.2 |
| 2 months | 2 | 18.2 |
| 3 months | 1 | 9.1 |
| 4 months | 6 | 54.5 |
| Ever heard of rubella disease | | |
| Yes | 9 | 6.2 |
| No | 136 | 93.8 |
| Knowledge assessment of rubella disease (N = 9) | | |
| Excellent | 1 | 11.1 |
| Good | 3 | 33.3 |
| Fair | 4 | 44.4 |
| Average | 1 | 11.1 |
| Vaccinated against rubella | | |
| Yes | 1 | 0.7 |
| I don't know | 23 | 15.9 |
| No | 121 | 83.4 |

SD = Standard deviation.

**Table 3. IgM, IgG seroprevalence and IgG avidity.**

| Characteristic | Frequency | Proportion (%) |
|---|---|---|
| Presence of IgM immunoglobulin | | |
| Positive | 5 | 3.5 |
| Negative | 130 | 89.6 |
| Indeterminate | 10 | 6.9 |
| Presence of IgG immunoglobulin | | |
| Positive | 133 | 91.7 |
| Negative | 12 | 8.3 |
| IgG avidity index | | |
| High | 128 | 96.2 |
| Low | 5 | 3.8 |

SD = Standard deviation.

There were ten clients with indeterminate IgM results. The average parity was 0.8 and the average gestational age was 32.6 weeks. Their ages ranged from 18–37 years. Sixty percent of this group were para 0 and twenty percent was para 1. The lowest gestational age at testing was twenty-four weeks.

Client 121 did an anomaly scan prior to taking part in the study which showed no obvious congenital abnormality. However, when the child was born, major congenital cardiac abnormalities were detected, for which plans were made for corrective surgery in South Africa. However, following the interruption of air travel due to the COVID pandemic, these plans did not materialise and child passed away soon after.

Client 132 delivered an apparently normal female baby at term. However, during follow up screening a hearing defect was detected in the right ear. The child is currently being followed up by the hearing assessment unit.

## Non-immune clients

Twelve clients were non-immune to rubella with no antibodies, which constitutes 8.28% of the study population. Those clients which could be contacted where directed to a specific health

**Table 4. Factors associated with IgG positivity in study participants.**

| Characteristic | IgG Status (n = 145) | | Odds ratio | p-value |
|---|---|---|---|---|
| | Positive | Negative | | |
| Age, years (mean ± SD) | 31.5 ± 5.6 | 34.2 ± 6.4 | 0.92 [0.83–1.02] | 0.123 |
| Parous | 102 (76.7) | 10 (83.3) | 1.52 [0.32–7.31] | 0.602 |
| Nulliparous | 31 (23.3) | 2 (16.7) | | |
| Number of children (mean ± SD) | 2.1 ± 1.2 | 2.0 ± 1.1 | 1.11 [0.62–1.98] | 0.726 |
| Type of residence | | | | |
| Peri-urban/Slum/Rural | 9 (6.7) | 0 | | 0.612* |
| Urban | 124(93.2) | 12 (100) | | |
| Educational level | | | | |
| JHS/Middle school and below | 53(39.8) | 4 (33.3) | 0.75(0.22–2.63) | 0.454* |
| SHS and above | 80(60.2) | 8(66.7) | | |

*Fisher's exact test; SD = Standard deviation; number (%).

**Table 5. Factors associated with IgM positivity and IgG avidity in study participants.**

| Characteristic | IgM Status (n = 135) | | | |
|---|---|---|---|---|
| | Positive (n =) | Negative (n =) | Odds ratio | p-value |
| Age, years (mean ± SD) | 30.4 ± 6.7 | 32.2 ± 5.5 | 0.94 [0.7 9–1.12] | 0.485 |
| Parity | | | | |
| Yes | 4 (80.0) | 104 (80.0) | 1.00 | |
| No | 1 (20.0) | 26 (20.0) | 1.00 [0.12–9.32] | 1.000* |
| Number of children (mean ± SD) | 1.8 ± 0.5 | 2.1 ± 1.2 | 0.73 [0.27–2.00] | 0.543 |
| Area of residence | | | | |
| Peri-Urban/Rural/Slum | 1 (20.0) | 6(4.6) | 5.17(0.50–53.61) | 0.237* |
| Urban | 4 (80.0) | 124 (95.4) | | |
| Educational level | | | | |
| JHS/Middle school and below | 3 (60.0) | 48(36.9) | 2.04 [0.32–12.86] | 0.447 |
| SHS and above | 2(40.0) | 82(63.1) | | |
| Characteristic | IgG avidity (n = 133) | | | |
| | High (n =) | Low (n =) | Odds ratio | p-value |
| Age, years (mean ± SD) | 31.6 ± 5.6 | 29.2 ± 3.3 | 1.08 [0.92–1.28] | 0.339 |
| Parity | | | | |
| Yes | 97 (75.8) | 5 (100) | 1.00 | |
| No | 31 (24.2) | 0 (0) | Not estimable | 0590 |
| Number of children (mean ± SD) | 2.1 ± 1.2 | 3.0 ± 1.6 | 0.59 [0.31–1.14] | 0.115 |
| Type of residence | | | | |
| Peri-urban/Slum/Rural | 9 (7.0) | | Not estimable | 0.70 |
| Urban | 119(93.0) | 5 (100) | | |
| Educational level | | | | |
| JHS/Middle school and below | 50 (39.1) | 3(60.0) | 0.43 (0.07–2.64) | 0.312 |
| SHS and above | 78 (60.9) | 2 (40.0) | | |

*Fisher's exact test; SD = Standard deviation; number (%); ten IgM indeterminate clients were excluded from the analysis.

**Table 6. Estimation of time of infection of pregnant women with rubella virus immunoglobulin M (IgM).**

| Clients with low avidity IgG | | | | | |
|---|---|---|---|---|---|
| Patient ID | Age | Parity | Avidity (%) | GA$^T$ | GA$^I$ |
| 040 | 30 | 2 | 38.4 | 35 w 3 d | 29 w |
| 079 | 35 | 3(1d) | 26.7 | 32 w 4 d | 26 w |
| 095 | 28 | 3 | 42.6 | N/A | N/A |
| 129 | 24 | 1 | 42.5 | 37 w 0 d | 31 w |
| 166 | 34 | 4 | 37.8 | 37 w 2 d | 31 w |
| **Clients with high avidity IgG** | | | | | |
| 036 | 32 | 2 | 80.5 | 36 w 4 d | 23 w |
| 100 | 21 | 0 | 89.7 | 31 w 1 d | 18 w |
| 115 | 36 | 2(1d) | 82.9 | 34 w 1 d | 21 w |
| 161 | 24 | 1 | 110.8 | 34 w 1 d | 21 w |
| 168 | 36 | 2 | 67.3 | 16 w 0 d | 3 w |

GA$^T$, gestational age at the time of testing; GA$^I$, estimated time of infection with rubella virus; N/A, not available; avidity <45% = low, >50% = high.

agency for vaccination. The average parity was 0.7 and the average gestational age was 32.6 weeks. Their ages ranged from 27–46 years. Twenty–five percent of this group were para 0, sixty percent were para 1 or 2, and only one patient (8.3%) was para 3. It was interesting to note that 3 clients were above 40 years (25%). Three clients were also nulliparous. Two of the clients had bad obstetric history. 090 had antepartum haemorrrhage at 37 weeks resulting in a still birth. 135 had a child with a ventriculo septal defect, which was managed conservatively. However, the child passed away at 8 months from complications of the VSD.

### Follow-up data for babies

All investigations and assessments were paid for by the investigator, except the ophthalmic investigation, which was free, so cost was not a factor in performing investigations. Parents declined laboratory investigations as they felt that their children were well and a needle prick was unnecessary. Parents were also not able or willing to bring their children for the follow up assessments due to the three weeks lock down from mid-March to early April and subsequently felt that their children would be exposed to the SARS-COV-2 virus either in public transport or at the hospital. Despite allowances for travel expenses, some parents felt the risk outweighed any benefit their child would derive from these assessments. One family lived about 50 kilometres from Accra and did not have their own means of transport.

Number 168 was of interest as she was only 16 weeks pregnant at the time the sample was taken. The RV infection may likely have occurred in the first trimester or just before she became pregnant. The anomaly scan at twenty weeks did not show any gross abnormality. The child began vomiting 8 hours after delivery and was admitted to the NICU on account of inability to feed. Neonatal jaundice was detected within the first 24 hours which was managed with phototherapy. A septic screen was done which did not reveal any focus of infection, though the C reactive protein was elevated. All assessments done after that were normal.

### Discussion

Majority (91.72%) of the clients had been exposed to rubella virus. Out of this number, 5 (6.87%) had IgM and IgG, 10 (7.52%) clients had borderline results for IgM. Twelve clients (8.28%) had not been exposed to rubella before.

The high rubella IgG seroprevalence suggested by the Ghana Health Service in 2013, is similar to the findings in this study of 91.72%. This suggests that there is a steady-state transmission of RV within the population over time. Out of the 5 clients who had both types of antibodies, all had high avidity IgG, which suggests that they may have had acute rubella infection at some stage in the pregnancy. All clients with borderline IgM results had high avidity for IgG antibodies. Five women had low avidity IgG. One child, whose mother had a borderline test for IgM, had a hearing defect, but no serological evidence of rubella infection. Two babies died within 2 months after delivery from major congenital abnormalities which are not part of the CRS.

Many studies have been done to find the number of antenatal clients who have been exposed to RV across the West African sub-region by looking at the levels of IgG. However, fewer have attempted to identify the presence of IgM and to do avidity testing in pregnant women. The value for IgG obtained in the study is similar to that obtained in a study done in 1998 in Kumasi, which was 92.6%, and in 2013, 92.3% [4]. In Eikwe in the Western Region of Ghana, 84.3% of antenatal clients were positive for IgG which was lower than the other values [18]. This may be because of the rural nature of that study [22], but not also ruling out some limited differences in testing assays. Studies across West Africa within the past decade have shown prevalence rates of between 68.5% to 93.1%. In Cameroon, the IgG prevalence was

94.4% [23], while in Nigeria the IgG prevalence was between 68.5% to 93.1% [7, 9, 24, 25]. In Burkina Faso [12] and Senegal [6], prevalence rates of 93.3% and 90.1% respectively, have been observed. This shows that in many countries in West Africa, most pregnant women have been exposed to RV and suggests community transmission of wild-type virus.

In general, the percentage of antenatal attendants with IgM is low in West Africa, but shows that there is a group of pregnant women always at risk for RV infection. The prevalence of IgM in this study was lower (3.4%) than what was seen in Kumasi (6.6%) in 2013 [19]. This observation was made in a city, while a slighter lower prevalence of 4.7% was observed in a rural setting [18]. Though not statistically evaluated, it is expected that prevalence rates will be less in rural than urban settings due to crowding [26]. Some studies in pregnant women from hospital settings in Nigeria and Cameroon have also shown low prevalence rates, 3.9, 5.0, 1.84, 1.25% [14, 23, 25, 27], while others have reported relatively high rates 38.8%, and 16.3%, and 10.1% [9, 28, 29]. The high prevalence rates may be accounted for by possible outbreaks which may have resulted in both new and re-infections [9], the possibility of false positives should however not be discounted. It is noteworthy to mention that in Ethiopia a 9.5% prevalence of RV IgM was seen before the introduction of rubella vaccination [30].

In this study there was no statistically significant association between immunoglobulin status and age, parity, area of residence, and educational level. There was also no statistical association between avidity and age, parity, area of residence and educational level. This could mean that there is steady transmission of the rubella virus in the community and that between the ages of 18 to 46 years, most women would have been exposed to the virus. This lack of statistical association with socio demographic factors was also seen in other studies in Ghana, Senegal, Nigeria, and Cameroon [4, 6, 9, 23], and makes targeted vaccination a challenge for at-risk groups not covered by the immunization schedule. This explains the reason for the mass vaccination campaigns in South America which eliminated RV [31].

Three babies whose mothers were negative for IgM and had a low avidity for IgG were available for assessment. Two of these babies were brought for full assessment. None of the babies had any abnormality and none of the pointers to CRS were found. From the time at which the samples were taken, which was late in the third trimester, infection may have taken place in the preceding 6 weeks. It could also be that the mothers had waning immunity against naturally acquired infection. At those gestational ages, it is unlikely that any congenital defect will occur as the incidence of defects is highest in the first trimester and reduces to 50% by 16 weeks [32]. As not all infections lead to disease [32], the chances of the foetus being affected is further reduced.

One baby had already been diagnosed with multiple congenital defects, some of which could not be clearly identified on obstetric ultrasound scan due to anhydramnios. The ultrasound was able to identify pulmonary hypoplasia, cardiomegaly and the presence of cystic masses in the abdomen which appeared to be separate from the kidneys. At delivery the child had hydrocephalus and a distended abdomen and passed away six hours later. The mother appeared traumatised and despite being seen by the clinical psychologist, broke all contact with Korle-Bu for about a month after which she responded to phone calls again. The multiplicity of abnormalities did not match the criteria for CRS, though the infection is likely to have been present.

In the group which had IgM with high avidity IgG antibodies, two of the children were able to complete all assessments. Another client brought the baby only for hearing assessment and declined any laboratory investigations. The hearing assessment was normal. No abnormalities were found in 161. The presence of high avidity antibodies signifies that the infection occurred at least 13 weeks before the sample was taken [15]. For one of them who was 34 weeks at the time the sample was taken, the infection may have occurred around 21 weeks, at which time

the incidence of congenital defects is reduced as compared to the first trimester [32]. RV infection may not result in any obvious abnormality. Since CRS is a progressive disease and the history of exposure is present [33], some of the children have been scheduled for follow up visits at the paediatric ophthalmology clinic and hearing assessment unit.

There were 10 clients with indeterminate RV IgM results who all had high IgG avidity. Some data was available for 6 of the clients. The indeterminate range can present some diagnostic challenges. It could mean that the infection is acute with the IgM antibodies waning and IgG steadily rising. A reinfection is also possible with the high avidity representing antibodies from a previous infection occurring over a year ago. IgM antibodies may persist for more than a year and their presence may signify an infection that occurred before the index pregnancy [26].

One of the babies was of interest as the mother did an anomaly scan at 20 weeks' gestation which did not show any obvious abnormality. If the first scenario is taken, and the pregnancy was 29 weeks at the time the test was done, infection may have occurred around 16 weeks. At this gestational age, infection can lead to defects, though there is a reduced risk and is more likely to be a hearing impairment [32, 34]. The cardiac abnormality is not part of the common defects associated with congenital rubella syndrome. Another baby had a hearing defect detected in the right ear at assessment at age 5 months. At the time the maternal sample was taken, she was 40 weeks 4 days pregnant. If there was a recent RV infection, it may have occurred around 27 weeks, which is beyond the gestation at which hearing defects are commonly seen [32, 34]. The child is being followed up at the hearing assessment centre.

From the foregoing discussions, it is very unlikely that any of the children had CRS in spite of the fact that they may have been exposed during the pregnancies. Even though this is not evidence of the absence of CRS, the presence of a significant number of non-immune and exposed women still highlights the need to extend vaccination beyond the Expanded Programmes of Immunizations (EPI) in Ghana.

The percentage of non-immune clients was 8.28%, which was higher than the study in Kumasi in 1998 (7.4%) and 2013 (7.7%) [4, 19]. This is expected as these women are not part of the targeted demography of Ghana's RV vaccination strategy. It was lower than the number of non-immune patients in Eikwe in 2013; 15.7% [18]. This is comparable with results from other studies in the West African sub region reporting a range between 5.0% to 9.9% [6, 12, 23]. In a study conducted in Zaria in Northern Nigeria, 5% of antenatal attendants were found to be non-immune [9]. In a recently published study from Nigeria [24], the percentage of non-immune antenatal attendants was 13.9%. There is therefore scientific evidence of the presence of RV non-immune pregnant women in populations in the West African sub-region and this warrants a vaccination strategy to prevent CRS among this group.

## Conclusion

Majority of the clients (91.7%) were immune to rubella and 8.3% were non-immune. Five women (3.8%) were positive for IgM. Most clients had high avidity antibodies but 5 (3.8%) had low avidity antibodies. It appears that most of the women were probably in their second trimester when they became exposed to the RV and this probably explains why no babies had CRS. The cardiac defects detected in the study babies were not part of the defects commonly seen in CRS. The child with the hearing impairment will require further investigation with RT-PCR to determine if it was caused by the RV. As there remain a number of pregnant women at risk of contracting rubella virus, all antenatal clients should be screened for rubella during pregnancy and non-immune clients offered vaccination after delivery. Also, the national immunization programme should be expanded to include rubella vaccination for non-immune teenagers and adults.

## Author Contributions

**Conceptualization:** Naa Baake Armah, Kwamena W. Sagoe.

**Data curation:** Naa Baake Armah, Mercy Nuamah, Alfred E. Yawson, Edmund T. Nartey, Vera A. Essuman, Nana-Akyaa Yao, Kenneth K. Baidoo, Jemima Anowa Fynn, Derrick Tetteh, Eva Gyamaa-Yeboah, Makafui Seshie, Isaac Boamah, Kobina Nkyekyer.

**Formal analysis:** Naa Baake Armah, Kwamena W. Sagoe, Alfred E. Yawson, Edmund T. Nartey, Derrick Tetteh, Kobina Nkyekyer.

**Investigation:** Naa Baake Armah, Vera A. Essuman, Nana-Akyaa Yao, Kenneth K. Baidoo, Jemima Anowa Fynn, Derrick Tetteh.

**Methodology:** Naa Baake Armah, Kwamena W. Sagoe, Alfred E. Yawson, Makafui Seshie, Isaac Boamah.

**Project administration:** Naa Baake Armah, Derrick Tetteh, Eva Gyamaa-Yeboah.

**Supervision:** Kwamena W. Sagoe, Mercy Nuamah, Kobina Nkyekyer.

**Validation:** Kwamena W. Sagoe, Mercy Nuamah, Kobina Nkyekyer.

**Writing – original draft:** Naa Baake Armah.

**Writing – review & editing:** Kwamena W. Sagoe, Mercy Nuamah, Alfred E. Yawson, Edmund T. Nartey, Vera A. Essuman, Nana-Akyaa Yao, Kenneth K. Baidoo, Jemima Anowa Fynn, Derrick Tetteh, Eva Gyamaa-Yeboah, Makafui Seshie, Isaac Boamah, Kobina Nkyekyer.

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
