## [Decision Letter · Decision Letter 0]

14 Nov 2022

PONE-D-22-04342Rubella virus IgM and IgG antibodies with avidity in pregnant women and outcomes at a tertiary facility in Ghana: a cross-sectional studyPLOS ONE

Dear Dr. Sagoe,

Thank you for submitting your manuscript to PLOS ONE. After careful consideration, we feel that it has merit but does not fully meet PLOS ONE’s publication criteria as it currently stands. Therefore, we invite you to submit a revised version of the manuscript that addresses the points raised below during the review process.

We look forward to receiving your revised manuscript.

Kind regards,

Ray Borrow, Ph.D., FRCPath

Academic Editor

PLOS ONE

Journal Requirements:

Reviewers' comments:

Reviewer's Responses to Questions

**Comments to the Author**

1. Is the manuscript technically sound, and do the data support the conclusions?

Reviewer #1: Yes

Reviewer #2: Partly

2. Has the statistical analysis been performed appropriately and rigorously? 

Reviewer #1: Yes

Reviewer #2: No

3. Have the authors made all data underlying the findings in their manuscript fully available?

Reviewer #1: Yes

Reviewer #2: Yes

4. Is the manuscript presented in an intelligible fashion and written in standard English?

Reviewer #1: Yes

Reviewer #2: Yes

5. Review Comments to the Author

Reviewer #1: The overall quality of the work is good. The authors have provided sufficient information to understand the topic of interest, explain the steps of the research process clearly, and provided a good discussion of the study results. I do not have further comments to submit and I believe this work is ready for publishing.

Please accept my best regards,

Reviewer #2: Introduction (page 6): second line- specify 9-14 ? months or years

Study design and population : last sentence of first paragraph (page 7) : questionnaires were administered ‘to’ obtain … add ‘to’.

Why were pregnant women in the first trimester of pregnancy not included ?

Add a sub heading of ‘sample size’

Reference formula used for sample size determination and give the name if possible.

Why the age limit of 45 years ?

Data and blood sample collection (page 8): nothing was described on blood sample collection. Title should be revised

Testing for rubella virus specific antibodies and avidity (page 9 : on the first line, precise if plasma or serum was used for analyses.

Detection of Congenital Rubella Syndrome : Is the national Cardiothoracic center part of the Korle Bu Teaching hospital ? if not include under study area.

Results : define exposed and non-immune in the first sentence.

Socio-demographic characteristics : What does number of children represent ? are they children living in the house during the current pregnancy or the number of children the participants had?

Tables are named 5.1, 5.2, 5.3…. it should be corrected to be Table 1, 2, 3, …

Why was avidity test done on all participants ?

Show avidity results of participants that were IgM posisitve

Table 5.5 indicates IgM positivity. How many participants were IgM positive ?

Non-immune clients : specify if rubella vaccine given to participants during pregnancy.

Follow-up data for babies : define SARS-COV-2

Discussion : page 28, first sentence not clear.

6. PLOS authors have the option to publish the peer review history of their article (what does this mean?). If published, this will include your full peer review and any attached files.

Reviewer #1: No

Reviewer #2: No

---

## [Author Response · Author response to Decision Letter 0]

13 Dec 2022

General responses:

We did not observe any statistical issues and all the ethical details have been provided.

Reviewer #2:

Comment: Introduction (page 6): second line- specify 9-14? months or years

Response: The word “years” has been inserted

Comment: Study design and population: last sentence of first paragraph (page 7): questionnaires were administered ‘to’ obtain … add ‘to’.

Response: The word “to” has been inserted

Comments: Why were pregnant women in the first trimester of pregnancy not included?

Response: There are no clear policies concerning termination of pregnancies in the event of indications that the foetus may have been infected with Rubella virus. This will lead to a dilemma for the Obstetricians and present challenges with patients in decision making.

Comment: Add a sub heading of ‘sample size’

Response: A sub-heading “Sample size and strategy” has been introduced above the formula for determining sample size.

Comment: Reference formula used for sample size determination and give the name if possible.

Response: The name of the formula has been given; Cochran formula is well known and no reference is required.

Comment: Why the age limit of 45 years?

Response: The study had a sampling strategy among pregnant women, and that was the maximum age sampled.

Comment: Data and blood sample collection (page 8): nothing was described on blood sample collection.

Response: Information on blood collection has been provided.

Comment: Title should be revised

Response: The phrase “a cross-sectional study” has been deleted

Comment: Testing for rubella virus specific antibodies and avidity (page 9: on the first line, precise if plasma or serum was used for analyses).

Response: The manuscript has already noted that plasma was used by obtaining blood using EDTA anticoagulant tubes.

Comment: Detection of Congenital Rubella Syndrome: Is the national Cardiothoracic center part of the Korle Bu Teaching hospital? if not include under study area.

Response: Yes, it is part of the Korle-Bu Teaching Hospital (https://kbth.gov.gh/departments-centres/)

Comment: Results: define exposed and non-immune in the first sentence.

Response: This has been done.

Comment: Socio-demographic characteristics: What does number of children represent? are they children living in the house during the current pregnancy or the number of children the participants had?

Response: The phrase “each participant had (parity)” has been included. These were the number of children the woman had delivered. 

Comment: Tables are named 5.1, 5.2, 5.3…. it should be corrected to be Table 1, 2, 3, …

Response: This has been done.

Comment: Why was avidity test done on all participants?

Response: To determine the recency of rubella virus IgG which was present in 92% of participants. This has been articulated in the introduction. Avidity testing is only done when IgG is detected.

Comment: Show avidity results of participants that were IgM posisitve

Response: This has been shown in Table 6.

Comment: Table 5.5 indicates IgM positivity. How many participants were IgM positive ?

Response: Five women as shown in Table 3

Comment: Non-immune clients: specify if rubella vaccine given to participants during pregnancy.

Response: It is unclear why this request is being made. None of the clients was vaccinated during pregnancy. Vaccination is contraindicated in pregnancy.

Comment: Follow-up data for babies: define SARS-COV-2

SARS-CoV-2 has been written in full, “Severe Acute Respiratory Syndrome Coronavirus Type 2”

Comment: Discussion: page 28, first sentence not clear.

Response: A slight modification has been made.

---

## [Editor Report · Decision Letter 1]

14 Dec 2022

Rubella virus IgM and IgG antibodies with avidity in pregnant women and outcomes at a tertiary facility in Ghana

PONE-D-22-04342R1

Dear Dr. Sagoe,

We’re pleased to inform you that your manuscript has been judged scientifically suitable for publication and will be formally accepted for publication once it meets all outstanding technical requirements.

Kind regards,

Ray Borrow, Ph.D., FRCPath

Academic Editor

PLOS ONE
---

## [Editor Report · Acceptance letter]

22 Dec 2022

PONE-D-22-04342R1 

Rubella virus IgM and IgG antibodies with avidity in pregnant women and outcomes at a tertiary facility in Ghana 

Dear Dr. Sagoe:

I'm pleased to inform you that your manuscript has been deemed suitable for publication in PLOS ONE. Congratulations! Your manuscript is now with our production department. 

Kind regards, 

on behalf of

Prof. Ray Borrow 

Academic Editor

PLOS ONE